# Automatic Classification of Cat Vocalizations Emitted in Different Contexts

**DOI:** 10.3390/ani9080543

**Published:** 2019-08-09

**Authors:** Stavros Ntalampiras, Luca Andrea Ludovico, Giorgio Presti, Emanuela Prato Previde, Monica Battini, Simona Cannas, Clara Palestrini, Silvana Mattiello

**Affiliations:** 1Department of Computer Science, University of Milan, 20133 Milan, Italy; 2Department of Pathophysiology and Transplantation, University of Milan, 20133 Milan, Italy; 3Department of Veterinary Medicine, University of Milan, 20133 Milan, Italy

**Keywords:** acoustic signal processing, pattern recognition, cat vocalizations

## Abstract

**Simple Summary:**

Cat vocalizations are their basic means of communication. They are particularly important in assessing their welfare status since they are indicative of information associated with the environment they were produced, the animal’s emotional state, etc. As such, this work proposes a fully automatic framework with the ability to process such vocalizations and reveal the context in which they were produced. To this end, we used suitable audio signal processing and pattern recognition algorithms. We recorded vocalizations from *Maine Coon* and *European Shorthair* breeds emitted in three different contexts, namely *waiting for food*, *isolation in unfamiliar environment*, and *brushing*. The obtained results are excellent, rendering the proposed framework particularly useful towards a better understanding of the acoustic communication between humans and cats.

**Abstract:**

Cats employ vocalizations for communicating information, thus their sounds can carry a wide range of meanings. Concerning vocalization, an aspect of increasing relevance directly connected with the welfare of such animals is its emotional interpretation and the recognition of the production context. To this end, this work presents a proof of concept facilitating the automatic analysis of cat vocalizations based on signal processing and pattern recognition techniques, aimed at demonstrating if the emission context can be identified by meowing vocalizations, even if recorded in sub-optimal conditions. We rely on a dataset including vocalizations of *Maine Coon* and *European Shorthair* breeds emitted in three different contexts: *waiting for food*, *isolation in unfamiliar environment*, and *brushing*. Towards capturing the emission context, we extract two sets of acoustic parameters, i.e., mel-frequency cepstral coefficients and temporal modulation features. Subsequently, these are modeled using a classification scheme based on a directed acyclic graph dividing the problem space. The experiments we conducted demonstrate the superiority of such a scheme over a series of generative and discriminative classification solutions. These results open up new perspectives for deepening our knowledge of acoustic communication between humans and cats and, in general, between humans and animals.

## 1. Introduction

Understanding the mechanisms that regulate communication between animals and humans is particularly important for companion animals such as dogs (*Canis familiaris*) and cats (*Felis silvestris catus*), who live in close contact with their human social partners depending on them for health, care and affection.

Nowadays, cats are one of the most widespread and beloved companion animals: they are ubiquitous, share their life with people and are perceived as social partners by their owners [1,2]. Despite this, very few specific studies have been carried out to understand the characteristics of cat vocalizations and the mechanisms of vocal communication of this species with humans [3,4,5], let alone their automatic analysis.

Interspecific communication, whether visual, tactile, acoustic, etc. plays a fundamental role in allowing information sharing [6]. Animals use acoustic communication to transmit information about several specific situations (e.g., alarm, reproductive and social status). Despite the differences in the sound-generating apparatus of different animals [3,7,8], sound patterns can be handled in a common manner. The similarity among the various sound recognition schemes comes from the fact that a sound source has a very distinctive and characteristic way to distribute its energy over time on its composing frequencies, which constitutes its so-called *spectral signature*. This spectral signature comprises a unique pattern that can be revealed and subsequently identified automatically by employing statistical pattern classification techniques.

Similar studies, using machine learning techniques, have been conducted on a number of diverse species, e.g., bats, dolphins, monkeys, dogs, elephants, and numerous bird species [9,10,11].

The vocal repertoire of domestic cats includes calls produced with the mouth closed (e.g., purrs and trills), sounds produced while the mouth is held open in one position (e.g., spitting or hissing), and calls produced while the mouth is open and gradually closed (i.e., meow or miaow) [12]. In [13], cat vocalizations are used in different intra- and inter-specific contexts to convey information about internal states and behavioral intentions and have been shown to differ acoustically depending on how the sound was produced. Most of the sounds in vertebrates, including cats, are produced by means of the myoelastic-aerodynamic (MEAD) mechanism; however, other mechanisms, such as the active muscolar contraction (AMC), are present in cats for the emission of several vocalizations, such as purrs [14].

Cat meows represent the most characteristic intra-specific vocalization of felids, either wild or domesticated [15]. They are also the most common type of vocalization used by domestic cats to communicate with humans [12], whereas undomesticated felids rarely meow to humans in adulthood [16]. Meows are therefore particularly interesting for investigating cat communication towards humans.

In this work, we wish to assess the following research questions: ‘Do cat vocalizations exhibit consistent patterns across their frequency content to the extent that emission contexts are distinguishable, and, if so, can we use generalized sound recognition technology to model them?’. Following the recent findings in the specific field, we used acoustic parameters able to capture characteristics associated with the context, i.e. mel-frequency cepstral coefficients and temporal modulation [17,18,19,20] modeled by means of both generative (having at their core the hidden Markov model technology) and discriminative (support vector machines [21] and echo state networks [22]) pattern recognition algorithms. Unlike deep learning techniques, which are typically characterized by low interpretability levels [23], we aimed at a comprehensive classification scheme potentially revealing useful information about the problem at hand. An experimental procedure was carried out on a dataset including recordings representing *Maine Coon* and *European Shorthair* breeds to answer the research questions mentioned above, and identify the methodology best addressing the specific problem. In a broader perspective, this ongoing project is aimed at defining if further semantic features (e.g., $f_{0}$ and/or *roughness*) are a means used by the animal to convey a message.

This work is organized as follows. Section 2 provides details regarding the recording protocol and the acquired audio samples. Section 3 describes the feature extraction mechanism along with a graph-based classifier. In Section 4, we explain the parameterization of the classification methodology as well as experimental results. Finally, Section 5 concludes this work.

## 2. Building the Dataset

This section describes the data acquisition protocol and the obtained dataset. After the end of the project, an extended version of the dataset will be made publicly available to encourage further research and experimental reproducibility.

### 2.1. Treatments

The cats recruited for the present study by the veterinarians of our research group included:10 adult *Maine Coon* cats (one intact male, three neutered male, three intact females and three neutered females) belonging to a single private owner and housed under the same conditions; and11 adult *European Shorthair* cats (one intact male, one neutered male, zero intact females and nine neutered females) belonging to different owners and housed under different conditions.

Accompanied by least one veterinarian, cats were repeatedly exposed to three different contexts that were hypothesized to stimulate the emission of meows:*Waiting for food*: The owner started the normal routine operations that precede food delivery in the home environment, and food was actually delivered 5 min after the start of these routine operations.*Isolation in unfamiliar environment*: Cats were transported by their owner, adopting the normal routine used to transport them for any other reason, to an unfamiliar environment (e.g., a room in a different apartment or an office, not far from their home environment). Transportation lasted less than 30 min and cats were allowed 30 min with their owners to recover from transport, before being isolated in the unfamiliar environment, where they stayed alone for a maximum of 5 min.*Brushing*: Cats were brushed by their owner in their home environment for a maximum of 5 min.

The typical vocalizations of a single exposure were composed of many repeated meows.

### 2.2. Data Acquisition

During the design phase of the experimental protocol, some technical and practical problems regarding the recording process had to be solved and several specification requirements had to be met.

First, the experiment required us to obtain vocalization recordings while minimizing potential influences caused by environmental sounds/noises. Moreover, the characteristics of the room (e.g., topological properties, reverberations, presence of furniture, etc.) should not affect the captured audio signals. Data acquisition could not be conducted in a controlled anechoic environment, since an unfamiliar space might have influenced the behavior of the cat in the *waiting for food* and *brushing* contexts. Finally, the relative position of microphones with respect to the sound source had to be fixed in terms of distance and angle, as many characteristics to measure in vocalizations depend on such parameters (see Section 3.1). Figure 1 shows representative spectrograms of samples coming from the three considered classes.

Thus, the idea was to capture sounds as close as possible to the source. We explored a number of solutions, including wearable devices directly attached to the back of the animal. Such experimental computing systems encompassed the entire recording, processing and storing chain within a miniaturized board. This approach proved not to be applicable to cats, since the perception of a device in contact with the fur once again could have produced biased behavioral results; moreover, the problem of capturing sound in proximity of the vocal apparatus remained.

Thus, we adopted a very small and lightweight microphone placed under the cat’s throat through a collar, an object with which the animal is already familiar, or can get accustomed after a short training period. When a cat was not used to wearing a collar, it was trained by the owner to wear it for some days before data collection, until no sign of discomfort (such as scratching the collar, shaking, lip liking, yawning, etc.) was reported [24,25,26,27,28]. This process was successful for all cats, enabling acquisition of real-world cat vocalizations. Since the microphone placement was the same for all cats, sound modification due to angle, distance and relative position to the mouth were consistent across the recordings, thus marginalizing the effects of these aspects (for example information such as pitch does not change, and features such as MFCCs may be biased by a constant factor across recordings).

Finally, one could argue that different situations can lead to different cat postures, which can influence sound production. We do not consider this issue as a bias, since we are interested in the vocalization regardless of how the cat produced it. A posture change is considered a legitimate and realistic effect of the situation. Moreover, if the recording quality were sufficient to catch such an aspect, it could be considered as a useful additional information.

Concerning the recording device, several Bluetooth headsets presented the desired characteristics in terms of dimensions, range, and recording quality. Regarding the latter aspect, it is worth underlining that Bluetooth microphones are usually low-budget devices packed with mono earphones and optimized for human voice detection. For example, the frequency range correctly acquired is typically very limited if compared to high-quality microphones. Concerning budget aspects, the recognition task can be performed on an entry-level computer.

The selected product was the *QCY Q26 Pro Mini Wireless Bluetooth Music Headset* (see Figure 2). Its technical specifications declare a microphone dynamic range of 98±3 dB. The signal is transmitted via Bluetooth with HFP [29] or HSP [30] profiles, thus relying on a logarithmic *a-law* quantization at a sampling frequency of 8 kHz. As a consequence, the actual range of frequencies we could rely on is 0–4 kHz. We expected that the fundamental frequency emitted by cats falls within the range, whereas some higher-frequency harmonics were likely to be cut. One of the goals of our experimentation was testing if vocalization classification can be performed even under these conditions, thus demonstrating that the useful information is contained within a narrow spectrum.

The size (15×10×23 mm) and weight (50 g) of the device were sufficiently small to be carried by a cat without significant behavioral implications. Interestingly, this microphone (see Figure 3) is a very low-budget solution: the whole headset, including an earphone, at the moment of writing, is sold for less than 20.

The adoption of the Bluetooth communication protocol greatly simplified the recording chain: it was possible to pair the headset to a smartphone equipped with Easy Voice Recorder PRO, a high-quality PCM audio recording applications. This aspect was fundamental to obtain recordings in a non-supervised environment from people without specific knowledge in the IT field, like cats’ owners and/or vets.

Non-meowing sound recordings were discarded (actually, only one instance), and the remaining sounds were cut so that each file contained a single vocalization, with 0.5 s of leading and trailing silence segments. The final corpus consisted of 448 files, organized as shown in Table 1. The total number of files referring to *Maine Coon* specimens is 196, and 252 for *European Shorthair* cats. The average length of each file is 1.82 s, with a variance of 0.37 s. It should be noted that about 1 s of each file contains only background noise. This portion of signal was automatically removed during the analysis, as explained in Section 3.

## 3. Species-Independent Recognition of Cat Emission Context

This section describes the proposed methodology achieving recognition of cat emission context. We relied on the basic assumption forming all modern generalized sound recognition solutions [7,31,32,33,34] stating that the sound sources distribute their energy across the different frequency bands in a unique way. Thus, our goal is to capture and subsequently model the specific distribution towards identifying it in novel incoming audio recordings. The next subsections, respectively, analyze the employed features and the classification mechanism.

### 3.1. Feature Extraction

This works exploited two feature sets for capturing the characteristics of cat sound events:amel-frequency cepstral coefficients; andbtemporal modulation features.

Before extracting these sets, we applied a statistical-model-based silence elimination algorithm described in [35] so that both the feature extraction and classification mechanisms elaborate solely on the structure of the available sound events.

#### 3.1.1. Mel-Frequency Cepstral Coefficients (MFCC)

For the derivation of the first feature set, 23 mel filter bank log-energies were utilized. The extraction method was as follows: Firstly, the short time Fourier transform was computed for every frame while its outcome was filtered using triangular mel-scale filterbank. Consecutively, we obtained the logarithm to adequately space the data. Finally, we exploited the energy compaction properties that the discrete cosine transform benefits to decorrelate and represent the majority of the frame-energy with just a few of its coefficients. Lastly, the most important twelve coefficients were kept and, in combination with frame’s energy, a thirteen-dimension vector was formed. It should be mentioned that the first, second, and third derivatives were appended to form the final feature vector. The processing stage was based on the openSMILE feature extraction tool [36].

#### 3.1.2. Temporal Modulation Features

A modulation-frequency analysis via the Fourier transform and filtering theory formed the basis of this set, as described in [37,38]. Such an analysis has demonstrated relevant potential in the field of generalized sound emotion recognition [20] motivating its inclusion in this work. Modulation filtering processes slow-varying envelopes of spectral bands coming from non-stationary signals without affecting the signal’s phase nor structure. We employed the publicly available implementation called Modulation Toolbox [39]. This set emphasizes the temporal modulation, while assigning high frequency values to the spectrum parts affecting the cochlea of the listener.

The modulation representation differs from a typical power spectrogram since it emerges from modeling the human cochlea, where the vibrations existing in the inner ear are converted into electrically encoded signals. The basilar membrane is stimulated by incoming sounds, whose response is associated to the stimulus’ frequency. The areas of the membrane are stimulated by components with sufficient difference in terms of frequency. In other words, the output of the cochlea may be divisible into frequency bands. Under this assumption, the short-time excitation energy of each channel is the output of the corresponding band. An important observation here is that a harmonic sound event will generate similar modulation patterns across all bands. This is exactly the advantage of the modulation spectrogram as such redundancy is not present in traditional types of spectra coming from harmonic sounds [40].

Representative samples of both feature sets corresponding to all classes (*waiting for food*, *isolation in unfamiliar environment*, and *brushing*) are depicted in Figure 4.

### 3.2. Pattern Recognition

The audio pattern recognition module of the proposed framework operates after the detection of cat vocalizations. It adopts the concept of Directed Acyclic Graph (DAG), i.e., a finite directed graph with no directed cycles [41,42]. In this way, the classification scheme can be represented as the graph G={N,L}, where $N=\{n_{1},\ldots ,n_{m}\}$ denotes the nodes, and $L=\{l_{1},\ldots ,l_{k}\}$ the links associating the nodes. A binary classification task was performed by each node *n* in *N* through a set of hidden Markov models (HMM), thus originating the DAG-HMM naming.

This methodology is suitable to the specifications of audio pattern recognition. One of the main advantages is the possibility to break any $C_{m}$-class classification problem to a series of two-class classification problems, without having to deal with all different classes simultaneously ($C_{m}$ is the number of classes). More specifically, the presented DAG-HMM is able to split any $C_{m}$-class problem into a list of binary classification ones.

Essentially, DAG technology generalizes the decision trees, with the advantage that redundancies existing in the tree structure are easily understood, a process that leads to the efficient merging of redundant paths. Moreover, DAGs are able to sort meaningfully the (classification) tasks before executing them. Importantly, such a property is associated with the performance achieved with respect to the overall task in a straightforward way [43].

Here, DAG-HMM consists in $C_{m}\left(C_{m}-1\right)/2$ nodes, each one responsible for carrying out a specific classification task. The existing arcs have a single orientation, while loops are not present. Thus, each node in *N* has either zero or two leaving connections.

Next, we detail how DAG-HMM is built and orderly executes the included tasks. Interestingly, we provide a meaningful solution to DAG-HMM’s *topological ordering*, i.e., the ordered execution of tasks which is inferred by an early indication of the separability of the involved classes.

#### 3.2.1. Topological Ordering of G

As seen during early experimentations, the DAG-HMM’s performance was directly related to the order in which the different classification tasks were carried out. More precisely, it would be beneficial to form G so that simple tasks are placed earlier than more difficult ones. This way, classes which might produce a large amount of misclassifications are removed during the early stages of G’s operation. Towards getting an indication of each task’s difficulty, we measured the performance of HMM classification mechanism on a validation set. Finally, tasks were ordered in descending classification rates meaning that tasks associated with high rates were placed on the top of G leaving tasks associated with lower rates to be conducted later.

The algorithm responsible for the topological ordering is given in Algorithm 1. Its inputs are the audio dataset ($C_{m}$), the maximum number of HMM states maxStates, and maximum number of Gaussian functions maxGaussians, while its output is the ordering of nodes in G (Line 1, Algorithm 1). Initially the dataset was divided into training and validation sets, denoted as TS and VS, respectively (Line 2, Algorithm 1). After initializing the task-based recognition rates (Line 3, Algorithm 1), the HMM model space was explored for identifying the HMM providing the highest recognition rate for each task in $C_{m}$ (Line 4, Algorithm 1). Subsequently, the vector with the recognition rates *r* was sorted in a descending order (Line 5, Algorithm 1) providing the indices to sort $C_{m}$ (Line 6, Algorithm 1) and, in turn, the topological order of G.

#### 3.2.2. DAG-HMM Operation

DAG-HMM operates as follows: Initially, the features of the novel audio signal are extracted and fed to the first node. The respective feature sequence is matched against the HMMs included in the node producing two log-likelihoods. These are compared and the graph operation continues along the path of the maximum log-likelihood until assigning a class to the unknown signal. The HMMs in each node are optimized (in terms of number of states and Gaussian components), focusing on the classification task carried out by each node. In other words, a given class may be modeled by HMMs of different parameters across G. Figure 5 demonstrates the DAG addressing the three-class problem of classifying cat vocalizations.

**Algorithm 1:** The algorithm for determining the topological ordering of G.
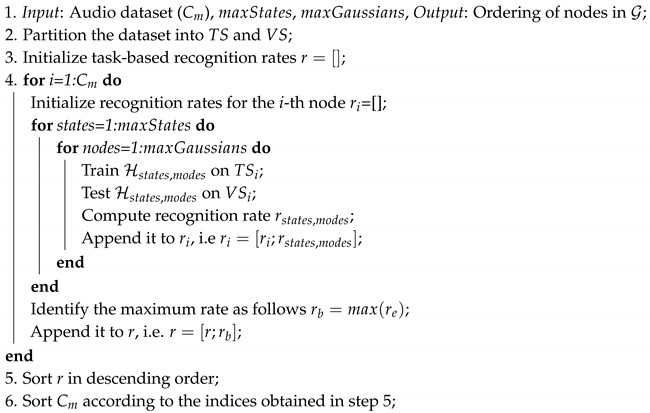


## 4. Experimental Set-Up and Results

This section provides thorough details regarding the parameterization of the proposed framework for classifying cat vocalizations, as well as the respective experimental results and how these compare with classification systems commonly used in the generalized sound classification literature [44], i.e., class-specific and universal HMMs, support vector machines, and echo state networks. The motivation behind these choices aimed at satisfying the condition of including both generative and discriminative pattern recognition schemes [45,46]. Moreover, during early experimentations, it was verified that MFCCs and temporal modulation features capture distinct properties of the structure of the available audio signals as they achieved different recognition rates characterized by diverse mislassifications, thus we decided to use them concurrently.

### 4.1. Parameterization

During the computation of the MFCC and TM features, we employed a frame of 30 ms with 10 ms overlap, so that the system is robust against possible misalignments. Moreover, to smooth any discontinuities, the sampled data were hamming windowed and the FFT size was 512.

Torch implementation (Torch is a machine-learning library publicly available at http://www.torch.ch) of GMM and HMM, written in C++, was used during the whole process. The maximum number of *k*-means iterations for initialization was 50 while both the EM and Baum–Welch algorithms [47] had an upper limit of 25 iterations with a threshold of 0.001 between subsequent iterations. In Algorithm 1, the number of states of the HMMs was selected from the set s∈{3,4,5,6} and the number of Gaussian functions from the set g∈{2,4,8,16,32,64,128,256} using the validation set alone.

Moving to support vector machine parameterization, its kernel function was a Gaussian radial basis, while the soft margin parameter and γ were determined by means of a grid search guided by cross-validation on the training set. The specific kernel was selected as it demonstrated superior performance with respect to other well-known ones, i.e., linear, polynomial homogeneous, pa refinement of the original idea wasolynomial inhomogeneous, and hyperbolic tangent.

The parameters of the ESN were selected by means of exhaustive search based on the minimum reconstruction error criterion using a validation set (20% of the training one). The parameters were taken from the following sets: spectral radius SR∈{0.8,0.9,0.95,0.99}, reservoir size G∈ {100, 500, 1000, 5000, 10,000, 20,000}, and scaling factor sc∈{0.1,0.5,0.7,0.95,0.99}. Its implementation was based on the Echo State Network Toolbox (the Echo State Network Toolbox for MATLAB is publicly available at http://reservoir-computing.org/software).

### 4.2. Experimental Results

Towards assessing the efficacy of the proposed classification mechanism, we followed the ten-fold cross validation experimental protocol. During this process, both training and testing parts were kept constant between different classifiers ensuring a fair comparison. The recognition rates corresponding to the considered approaches (DAG-HMM, class-specific HMMs, universal HMM, SVM and ESN) are tabulated in Table 2. The corresponding achieved rates are 95.94%, 80.95%, 76.19%, 78.51%, and 68.9%.

As we can see, the DAG-based classification scheme provides an almost perfect recognition rate outperforming the rest of the approaches. The second highest rate is offered by the class-specific HMMs, while the ESN achieves the lowest one. We argue that limiting the problem space by means of a graph structure is significantly beneficial in the problem of classifying cat vocalizations. Even though such an approach may be computationally expensive when the number of classes is very large, in problems with a reasonable amount of classes, it may able to provide encouraging results.

Table 3 presents the confusion matrix associated with the best-performing approach, i.e., DAG-HMM. We observe that the state *waiting for food* is perfectly recognized (100%, $F_{1}=0.98$). The second best recognized class is *brushing* (95.24%, $F_{1}=0.94$) and the third one is *isolation in unfamiliar environment* (92.59%, $F_{1}=0.96$). The misclassifications concern the classes *isolation* and *brushing* which are wrongly recognized as *brushing* and *waiting for food*, respectively. After inspecting the misclassified recordings, it was observed that the specific samples are acoustically similar (to the extent that a human listener can assess) with samples existing in the wrongly predicted classes.

It is worth noticing that the context *waiting for food* was never attributed to the other contexts, as it probably had characteristics that are specific of a positive state. The misclassifications occurred mainly for *isolation*, which was attributed to another context with an expected dominant negative valence, i.e., *brushing*. However, some vocalizations that occurred during *brushing* were attributed to the positive context *waiting for food*, suggesting that *brushing* can actually represent a positive stimulus for some cats.

#### Comment on DAG-HMM Applicability

At first sight, DAG generation could seem restrictive as it demands the construction of $\frac{N(N-1)}{2}$ nodes, a $O(n^{2})$ problem. However it is common that in practical audio classification applications, one needs to process a set including 10–20 classes, i.e., 45–190 nodes, which is not prohibitive given the advancements of modern information processing systems. Finally, model learning is to be conducted only once and offline, while during the classification process, one needs only to execute repetitive Viterbi algorithms, a process which is not computationally intensive [47].

## 5. Conclusions and Discussion

This work presents a solution for the task of classifying cat emission context on the basis of their vocalizations. Our aim was to use meaningful acoustic descriptors as well as classification solutions, such that, when a misclassification occurs, one is able to “open” the classification mechanism and get insights on how a given limitation may be surpassed. A highlight of the system is its ability to operate in such an imbalanced dataset in terms of both breed and sex. Moreover, the method presented here is generic and it can be applied with slight modifications to other vocalization-classification problems as long as the respective data become available.

Sounds were captured by a very low-budget recording device, namely a common Bluetooth headset, and analyzed using an entry-level computer. Despite its technical limitations, the system proved to be accurate enough in recording cat vocalizations. The reason may go back to an evolutionary explanation of cat behavior: since meows are basically produced to communicate with humans [15], we can expect that these sounds are close to the frequency range of human speech, which is quite broad depending on the sex and age of the speaker; this is exactly the frequency range where Bluetooth headsets are designed to achieve their best performances.

These results are quite encouraging and open up new perspectives for deepening our knowledge of acoustic communication between humans and cats and, in general, between humans and animals.

In our future work, we intend to deepen our understanding of cats’ emotional states in these three situations by complementing the proposed framework with behavioral data collected by video recordings. To this end, we intend to explore the following directions:compare emotional state predictions made by humans with those made by an automatic methodology;explore the sound characteristics conveying the meaning of cats’ meowing by analyzing the performance in a feature-wise manner;quantify the recognition rate achieved by experienced people working with cats when only the acoustic emission is available;establish analytical models explaining each emotional state by means of physically- or ethologically-motivated features; anddevelop a module able to deal with potential non-stationarities, for example new unknown emotional states.

## 6. Ethical Statement

The present project was approved by the Organism for Animals Welfare of the University of Milan (approval No. OPBA_25_2017). The challenging situations to which cats were exposed were required in order to stimulate the emission of meows related to specific contexts. They were conceived considering potentially stressful situations that may occur in cats’ lives and to which cats can usually easily adapt. To minimize possible stress reactions, preliminary information on the normal husbandry practices (e.g., brushing or transportation) to which the experimental cats were submitted and on their normal reactions to these practices were collected by interviews with the owners. Additionally, cats were video recorded using a camera connected to a monitor for 5 min before the stimulus, during the stimulus and for 5 min after the stimulus, in order to monitor their behavior during the isolation challenge, with the idea of stopping the experiment if they showed signs of excessive stress; however, such a situation never occurred.

## Figures and Tables

**Figure 1 animals-09-00543-f001:**
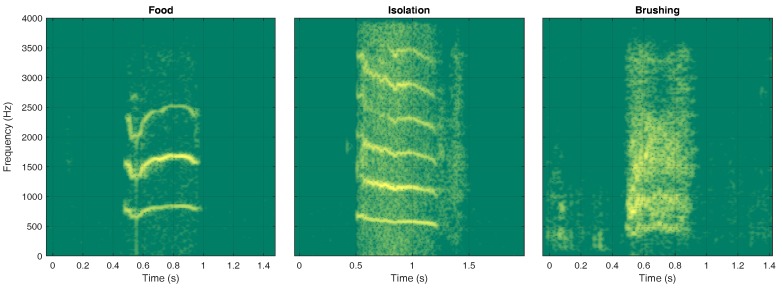
Time-frequency spectrograms of meows coming from the three considered classes.

**Figure 2 animals-09-00543-f002:**
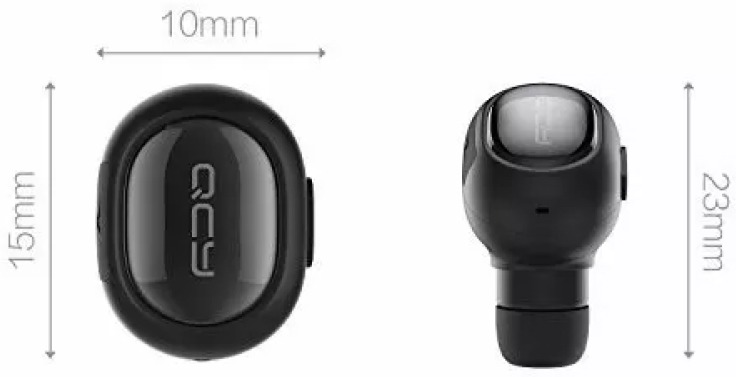
Details of the *QCY Q26 Pro Mini Wireless Bluetooth Music Headset*. The small hole in the middle of the right view is the microphone.

**Figure 3 animals-09-00543-f003:**
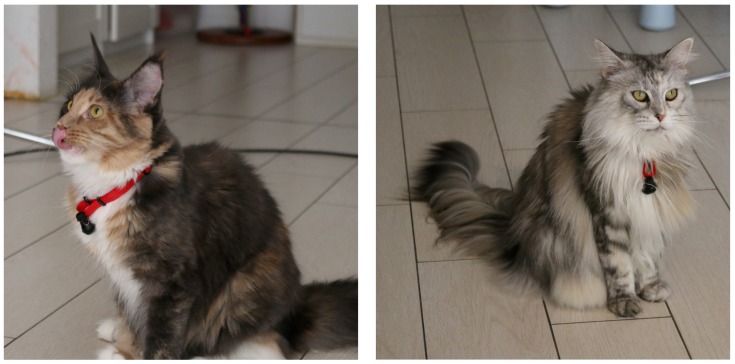
Two cats provided with a Bluetooth microphone placed on the collar and pointing upwards.

**Figure 4 animals-09-00543-f004:**
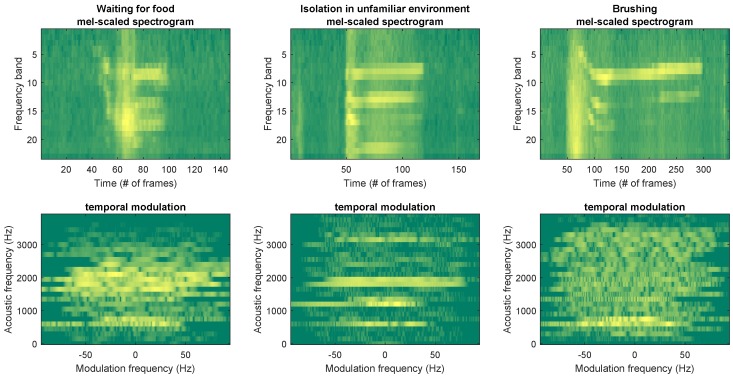
Representative representations of Mel-scaled spectrogram and temporal modulation corresponding to all three classes (*waiting for food*, *isolation in unfamiliar environment*, and *brushing*).

**Figure 5 animals-09-00543-f005:**
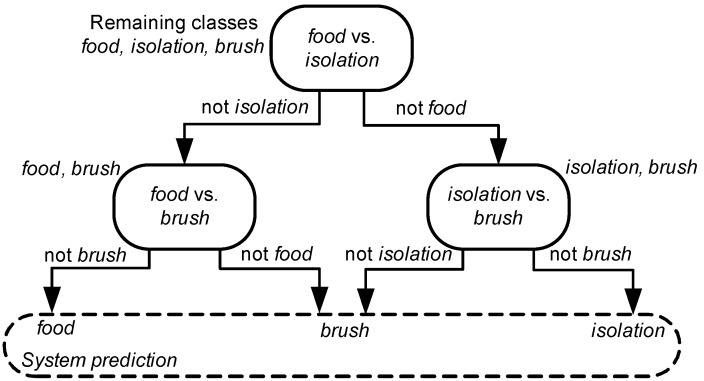
The DAG-HMM addressing the three-class problem of classifying cat manifestations. At each level, the remaining classes for testing are mentioned beside each node. Digging inside each node, an HMM-based sound classifier is responsible for activating the path of the maximum log-likelihood.

**Table 1 animals-09-00543-t001:** Dataset composition (MC, Maine Coon; ES, European Shorthair; IM/IF, Intact Males/Females; NM/NF, Neutered Males/Females).

	Food (93)		Isolation (220)		Brushing (135)	
	MC (40)	ES (53)	MC (91)	ES (129)	MC (65)	ES (70)
IM (20)	-	5	10	-	5	-
NM (79)	14	8	17	15	21	4
IF (70)	22	-	28	-	20	-
NF (279)	4	40	36	114	19	66

**Table 2 animals-09-00543-t002:** The recognition rates achieved by each classification approach. The highest one is emboldened.

Classification Approach	Recognition Rate (%)
Directed acyclic graphs—Hidden Markov Models	**95.94**
Class-specific Hidden Markov Models	80.95
Universal Hidden Markov Models	76.19
Support vector machine	78.51
Echo state network	68.9

**Table 3 animals-09-00543-t003:** The confusion matrix (in %) representing the classification results achieved by the DAG-HMM.

	Responded	*Waiting for Food*	*Isolation*	*Brushing*
Presented				
*Waiting for food*		**100**	-	-
*Isolation*		-	**92.59**	7.41
*Brushing*		4.76	-	**95.24**

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
