# Peer review of "Automatic Classification of Cat Vocalizations Emitted in Different Contexts"

_animals, 2019, doi:10.3390/ani9080543_

Round 1

Reviewer 1 Report

The paper presents a method to automatically analyze cat vocalizations emitted in different contexts using a DAG-HMM as classifier. The authors also describe the interesting data acquisition procedure to build the dataset used in their experiments. However, I see a few issues that should be resolved before publishing this paper:

1.     My main concern regarding this paper is precisely the dataset which has not yet been made public and is not provided as supplementary material. Therefore, it is very difficult to establish the significance of its results. I would like to encourage authors to provide the dataset as supplementary material to the scientific community to enable the evaluation and reproduction of their work.

2.     There is a relevant study in the literature that should be cited and compared with the method applied in this work to evaluate their strengths and weaknesses and taken into consideration in the conclusions and discussion of the work presented.:

Pandeya, Y.R.; Kim, D.; Lee, J. Domestic Cat Sound Classification Using Learned Features from Deep Neural Nets. Appl. Sci. 2018, 8, 1949.

3.     It seems that the recordings have been manually cut so that each file contains a single vocalization simplifying the preprocessing of the signal by removing most of the background noise. In most animal sound classification problems, the recordings contain several vocalizations of the same animal with background noise and the algorithm must discern between the calls of that animal. So, the method presented in this work cannot be considered completely automatic. This aspect should be clarified in the paper.

4.     Feature extraction should be better explained. Specially the temporary modulation features. It only refers to the toolbox used, but there is no clear mention of the demodulation method used or its parameterization (number of harmonics, etc.) or the size of the features extracted by each sound. I have tried to check the toolbox and it is no longer available on the referenced web page, the links are broken.

5.     In the experimental results it is mentioned that ten-fold cross validation has been used but it does not seem that any type of cross validation has been used to initialize the number of states and the number of Gaussian functions of the DAG-HMM. According to Alg. 1 it appears that the complete dataset with an undefined split has been used. This point should be clarified.

6.      The results should be presented in terms of F1-score, which are more appropriate if the distribution of classes is uneven.

7.     It would be very interesting for the reader that the features should be analyzed individually in the experimentation section to analyze their contribution to the result. So that it can be verified whether their contribution is significant.

8.     Several toolboxes or software implementations are mentioned in the paper that must be properly cited and not only indicate the web where it can be obtained

Minor comments:

1.     A minor issue is that the text in general needs polishing. I have found some misspelled words and improper use of abbreviations.

2.     Figure 3 is unclear. The image must be improved to allow an optimal interpretation. The units of some of the axes are also missing.

Author Response

Thank you very much for the opportunity to improve the paper. We would like to thank the Editor and the Reviewers for their constructive comments.

Comments from the Reviewer:

1) My main concern regarding this paper is precisely the dataset which has not yet been made public and is not provided as supplementary material. Therefore, it is very difficult to establish the significance of its results. I would like to encourage authors to provide the dataset as supplementary material to the scientific community to enable the evaluation and reproduction of their work.

2) There is a relevant study in the literature that should be cited and compared with the method applied in this work to evaluate their strengths and weaknesses and taken into consideration in the conclusions and discussion of the work presented.

3) It seems that the recordings have been manually cut so that each file contains a single vocalization simplifying the preprocessing of the signal by removing most of the background noise. In most animal sound classification problems, the recordings contain several vocalizations of the same animal with background noise and the algorithm must discern between the calls of that animal. So, the method presented in this work cannot be considered completely automatic. This aspect should be clarified in the paper.

4) Feature extraction should be better explained. Specially the temporary modulation features. It only refers to the toolbox used, but there is no clear mention of the demodulation method used or its parameterization (number of harmonics, etc.) or the size of the features extracted by each sound. I have tried to check the toolbox and it is no longer available on the referenced web page, the links are broken.

5) In the experimental results it is mentioned that ten-fold cross validation has been used but it does not seem that any type of cross validation has been used to initialize the number of states and the number of Gaussian functions of the DAG-HMM. According to Alg. 1 it appears that the complete dataset with an undefined split has been used. This point should be clarified.

6) The results should be presented in terms of F1-score, which are more appropriate if the distribution of classes is uneven.

7) It would be very interesting for the reader that the features should be analyzed individually in the experimentation section to analyze their contribution to the result. So that it can be verified whether their contribution is significant.

8) Several toolboxes or software implementations are mentioned in the paper that must be properly cited and not only indicate the web where it can be obtained.

Response of the author:

Thank you very much for your comment.

- As per the schedule of the project, the dataset will be published after its conclusion.

- The suggested reference is already included in the paper. In this work, we are not interested in black-box type of solutions, since we have to provide clear insights to veterinarians regarding the decision of the system. To clarify this, the following text is in the paper: “Our aim was to use meaningful acoustic descriptors as well as classification solutions, such that when a misclassification occurs, one is able to “open” the classification mechanism and get insights on how a given limitation may be surpassed.”

- The paper is not focused on the detection of cat vocalizations but on their recognition. We better framed our scopes by adding the following sentence: “The audio pattern recognition module of the proposed framework operates after the detection of cat vocalizations.”

- We retrieved the new URL for the Modulation Toolbox, and all the details are provided there.

- We have clarified this point in the sentence: “In Alg. 1, the number of states of the HMMs was selected from the set […] using the validation set alone.”

- F1 scores were added in the section including the experimental results.

- Feature-wise analysis is going to be addressed in our future work. We specified it more clearly by adding an explicit additional goal: “We intend to explore the following directions: […] explore the sound characteristics conveying the meaning of cats' meowing by analysing the performance in a feature-wise manner”

- We provided more information in the footnotes and the adequate references where available.

- All the typos were corrected.

Reviewer 2 Report

This paper describes a potentially useful application for the automatic classification of animal sounds by context, but in its current format it is difficult for even an established practitioner of bioacoustics to be able to understand, let alone those concerned about animal well-being who may also be interested in the application. To be maximally useful, the paper must be much clearer and less jargony in its presentation. I would also like to see a stronger case made for the successful validation of the new method.

One of the points that is noteworthy is that what appears to be the same vocalization (meow) is different in the three contexts that were tested. It would be helpful to display time-frequency spectrograms of meows from each context to illustrate how the new method might improve on more traditional methods. Many recent studies in nonhuman primates have found calls that to human ears sound identical also have subtle variants that vary with context, so this is interesting from the perspective of cognition in cats.  

The current paper tests the new method against four other methods and finds a better accuracy with the new method. But I was unclear on what the other four methods did. Even if they are described in other publications, a reader here needs to understand the differences among the classification methods and perhaps some examples of ways in which the other methods have been used in the past. This is important if one is to evaluate the success of the new method.

Also, the authors say that this method is likely to be useful to shelters, veterinarians and others, but these are areas that are chronically underfunded so the paper also needs a frank discussion of the costs involved including the cost of professional expertise needed to make the system work effectively. Ideally, this paper and supplemental materials might provide a map for how underfunded agencies could actually afford to use this method. The use of inexpensive blue-tooth enabled microphones is great but only if the analyses can be equally inexpensive.

Now to some more specific concerns:

l. 65 explain what Mel-Frequency cepstral coefficients are and give the acronym that will be used later in the paper to denote these.

l. 66 what is Hidden Markov model technology? Some readers may be familiar with Markov chains but not this.

l. 67 what are support vector machines and echo state networks?

l. 74 “a means”

l. 106 “required us”

l. 122 “used to wearing a collar”

l. 128 MFCC’s are not defined. The acronym did not appear when the term was spelled out. One cannot expect readers to understand this.

l. 157 define pre- and post-roll

l. 176 “cat-produced sound events” the idea hear is to cancel out ambient noise if I understand this.

l. 177 this entire section needs more clarity for readers naïve to the physics of sound analysis.

l. 186 what exactly does the OpenSMILE feature extraction tool do?

l. 197 is the human basilar membrane suitable for modeling cat vocalizations. Why not model on the cat basilar membrane?

l. 202 “similar” or “identical” rather than “akin”

l. 207 Unpack what is meant by Direct Acyclic Graph logic and put the acronym in parentheses after the term.

l. 216 Cm-class has not been defined anywhere yet. What does it refer to?

l. 234 “As was seen…performance was…to the order”

ll. 234-241 all verbs in this paragraph should be in the past tense.

l. 245 “dataset was divided”

l. 250-258 This might be better place in supplemental materials.

Algorithm 1 What is the cardinality operator? Where does it appear in the operation sequence?

Fig. 4 legend “the remaining classes”

Table 2 as noted earlier a reader even one with some background in bioacoustics needs help in understanding the differences between the methods evaluated: Class-specific Hidden Markov Models, Universal Hidden Markov Models, Support Vector Network, Echo State Network. Are any of these related to the more traditional Principal Components and Discriminative Analyses that have been more traditionally used?

Section 4.1 Parameterization was complete obscure to this reader.

l. 270 what is Torch Implementation?

l. 272 what are Baum Welch algorithms?

l. 290 “almost perfect recognition” 96% is already excellent by any bioacoustic standards.

The other models tested also do quite well relative to the chance rate of 33% so what is the cost of perfection?

l. 319 “experimented with before”

l. 322 neither breed nor sex have been explicitly tested in this paper so there may indeed be differences that are obscured by putting all the data together. “Gender” is typically thought to refer to one’s sexual identification, regardless of biological sex and in the absence of knowledge of how each cat thinks about its gender identity the word “sex” is the more appropriate term.  

l. 327 Examination of traditional spectrograms of cat vocalizations suggest that on average their calls are about an octave higher than human speech, but still within the 4kHz range of the microphones. It is more accurate to write that their meow calls are within the sensitive range of the microphones.

l. 334 Why was not a simple test of human predictability based on traditional sound spectrograms compared with this new method done for this paper? Such a comparison, if successful, would add considerable strength to the argument that the new method is superior.

I’d also like to see a discussion of how readily this methodology could be applied to other cat vocalizations- hisses, purrs, etc. Is it only really useful for sorting the contexts of meows?  

Author Response

Thank you very much for the opportunity to improve the paper. We would like to thank the Editor and the Reviewers for their constructive comments.

Comments from the Reviewer:

1) It would be helpful to display time-frequency spectrograms of meows from each context to illustrate how the new method might improve on more traditional methods.

2) The use of inexpensive blue-tooth enabled microphones is great but only if the analyses can be equally inexpensive.

3) explain what Mel-Frequency cepstral coefficients are and give the acronym that will be used later in the paper to denote these.

4 and 5) What is Hidden Markov model technology? Some readers may be familiar with Markov chains but not this. What are support vector machines and echo state networks?

6) Define pre- and post-roll

7) Table 2 as noted earlier a reader even one with some background in bioacoustics needs help in understanding the differences between the methods evaluated: Class-specific Hidden Markov Models, Universal Hidden Markov Models, Support Vector Network, Echo State Network. Are any of these related to the more traditional Principal Components and Discriminative Analyses that have been more traditionally used?

8) What is Torch Implementation? What are Baum Welch algorithms?

9) Examination of traditional spectrograms of cat vocalizations suggest that on average their calls are about an octave higher than human speech, but still within the 4kHz range of the microphones. It is more accurate to write that their meow calls are within the sensitive range of the microphones.

10) I’d also like to see a discussion of how readily this methodology could be applied to other cat vocalizationshisses, purrs, etc. Is it only really useful for sorting the contexts of meows?

Response of the author:

Thank you very much for your comment.

- we added the requested image (Figure 2) with three representative samples.

- on line 325, the text now is “Sounds were captured by a very low-budget recording device, namely a common Bluetooth headset, and analyzed using an entry-level computer.”

- on page 180, there is a definition about MFCCs: “For the derivation of the first feature set, 23 Mel filter bank logenergies are utilized. The extraction method is the following: Firstly, the short time Fourier transform is computed for every frame while its outcome is filtered using triangular Mel-scale filterbank. Consecutively we obtain the logarithm to adequately space the data. Finally, we exploit the energy compaction properties that the Discrete Cosine transform benefits in order to decorrelate and represent the majority of the frame-energy with just a few of its coefficients. Lastly, the most important twelve coefficients are kept and, in combination with frame’s energy, a thirteen-dimension vector is formed. It should be mentioned that the first, second, and third derivatives were appended to form the final feature vector. The processing stage was based on the openSMILE feature extraction tool [34]”.

- Explaining all the classifiers in detail goes beyond the scope of the paper, and some of the references with the requested definitions are already cited: [30], [46]. Anyway, we added two references after the first occurrence of SVM and ESN: [21], [22].

- We modified the terms into “leading and trailing silence segments”.

- These methods are very well-known in the Computer Science field, so describing them here is beyond our scope. We provided appropriate references for the interested reader.

- We added a reference and modified the footnote in order to clarify these concepts.

- We better explained such a concept specifying that cat meows are close to the frequency range of human speech, not human hearing: “The reason may go back to an evolutionary explanation of cat behavior: since meows are basically produced to communicate with humans [15], we can expect that these sounds are close to the frequency range of human speech, which is quite broad depending on the sex and age of the speaker; this is exactly the frequency range where Bluetooth headsets are designed to achieve their best performances.”

- In the conclusion we mentioned the possibility to generalize the method to other vocalization-classification problems.

- All the typos were corrected.

Round 2

Reviewer 1 Report

The authors have improved the paper, however, some of my previous comments were not addressed adequately.

 I still miss a proper comparison with the work of Pandeya et al. to see the strengths and weaknesses regarding this work. In addition. in the conclusions (line 318), the sentence " To the best of our knowledge, an approach based on a computational intelligence methodology was never experimented with before " should be removed or modified as there is prior knowledge of other intelligent systems applied to cat speech recognition.

In my question number seven, it seems my comment was misinterpreted. I mean to analyze the individual contribution of the two sets of features used, MFCCs and Temporal Modulation Features, to verify if their combination is appropriate.

The Modulation Toolbox has not been properly cited:

Les Atlas, Pascal Clark and Steven Schimmel, Modulation Toolbox Version 2.1 for MATLAB, https://sites.google.com/a/uw.edu/isdl/projects/modulation-toolbox, University of Washington, September 2010

Author Response

Reviewer's comments: The authors have improved the paper, however, some of my previous comments were not addressed adequately. 

I still miss a proper comparison with the work of Pandeya et al. to see the strengths and weaknesses regarding this work. In addition. in the conclusions (line 318), the sentence " To the best of our knowledge, an approach based on a computational intelligence methodology was never experimented with before " should be removed or modified as there is prior knowledge of other intelligent systems applied to cat speech recognition.

In my question number seven, it seems my comment was misinterpreted. I mean to analyze the individual contribution of the two sets of features used, MFCCs and Temporal Modulation Features, to verify if their combination is appropriate.

The Modulation Toolbox has not been properly cited:

Les Atlas, Pascal Clark and Steven Schimmel, Modulation Toolbox Version 2.1 for MATLAB, https://sites.google.com/a/uw.edu/isdl/projects/modulation-toolbox, University of Washington, September 2010

Response: Thank you very much for these suggestions.  All comments your bought up were implemented as you see in the manuscript. We would like to thank the Editor and the Reviewers for their constructive comments.

Reviewer 2 Report

The authors have made some minimal changes in response to m previous review, but have ignored many other points that I raised. In response, they state that these techniques are familiar ones in computer science and do not need to be explained in any depth here. However, this is a journal for biological scientists and not computer scientists so if they wish to publish for biologists, they need to provide more explanation.

Terms are somewhat better defined and I find the traditional frequency X time displays helpful. I still have trouble relating the terms used to describe the other methods to methods with which I am familiar as a bioacoustician. The authors have not addressed the issue I raised about the practicality and low cost of this method for veterinarians shelter managers. It appears that one can only use this method in conjunction with a team of engineers and computer scientists. Is the somewhat improved success of this method 96% correct versus approximately 80% for the next three methods worth the extra time and money? How successful are people experienced in working with cats in discriminating the call contexts by ear alone? This is a necessary test to determine the utility of this method.

Minor points:

l. 90 “hypothesized” not “supposed”

l. 133 “legitimate” instead of “legit”

l. 276 “were” not “where”

l. 278 “with respect to” rather than “wrt”

l. 281-283 SR, G and sc are not defined.

l. 290 “almost excellent” as noted in the previous review, this result IS excellent. I’d say “almost perfect”

l. 310 “first sight” not “first site”

Reference section has a mix of styles in terms of capitalizing words in titles. These should be consistent with the journal’s style.

Author Response

Thank you very much for these suggestions.  All comments were implemented as you see below. We would like to thank the Editor and the Reviewers for their constructive comments.

1) The authors have made some minimal changes in response to m previous review, but have ignored many other points that I raised. In response, they state that these techniques are familiar ones in computer science and do not need to be explained in any depth here. However, this is a journal for biological scientists and not computer scientists so if they wish to publish for biologists, they need to provide more explanation.

Responses: Thank you for this comment. The analysis of the classification approaches used in this work would not provide any type of added value to the manuscript. The interested reader including biological scientists can easily find all necessary details in the provided references. The DAG-based classifier is explained in detail as it comprises a novel classification paradigm.

We put emphasis on the motivation behind the choice of the included classification approaches -

- in the Abstract, line 18, the text is:

“Towards capturing the emission context, we extract two sets of acoustic parameters, i.e. Mel-Frequency Cepstral Coefficients and Temporal Modulation features. Subsequently, these are modeled using a classification scheme based on a directed acyclic graph dividing the problem space. The experiments we conducted demonstrate the superiority of such a scheme over a series of generative and discriminative classification solutions.”

- in line 64, the text is:

“Following the recent findings in the specific field, we used acoustic parameters able to capture characteristics associated with the context, i.e. Mel-Frequency cepstral coefficients and temporal modulation [17–20] modeled by means of both generative (having at their core the hidden Markov model technology) and discriminative (support vector machines [21] and echo state networks [22]) pattern recognition algorithms.”

- in line 259, the text is:

“This section provides thorough details regarding the parameterization of the proposed framework for classifying cat vocalizations, as well as the respective experimental results and how these compare with classification systems commonly used in the generalized sound classification literature [43], i.e. class-specific and universal HMMs, support vector machines, and echo state networks. The motivation behind these choices aims at satisfying the condition of including both generative and discriminative pattern recognition schemes [44,45].”

2) Terms are somewhat better defined and I find the traditional frequency X time displays helpful. I still have trouble relating the terms used to describe the other methods to methods with which I am familiar as a bioacoustician. The authors have not addressed the issue I raised about the practicality and low cost of this method for veterinarians shelter managers. It appears that one can only use this method in conjunction with a team of engineers and computer scientists. Is the somewhat improved success of this method 96% correct versus approximately 80% for the next three methods worth the extra time and money? How successful are people experienced in working with cats in discriminating the call contexts by ear alone? This is a necessary test to determine the utility of this method.

Response: Thank you very much for your comment. Regarding the cost of the device we made the following changes:

- on line 136, the text now is

“Concerning the recording device, several Bluetooth headsets presented the desired characteristics in terms of dimensions, range, and recording quality. Regarding the latter aspect, it is worth underlining that Bluetooth microphones are usually low-budget devices packed with mono earphones and optimized for human voice detection. For example, the frequency range correctly acquired is typically very limited if compared to high-quality microphones. Concerning budget aspects, the recognition task can be performed on an entry-level computer.”

- on line 328, the text now is

“Sounds were captured by a very low-budget recording device, namely a common Bluetooth headset, and analyzed using an entry-level computer.”

The execution method does not necessarily require a team of engineers and computer scientists but rather average computer skills. However, its training and optimization require such a team. Finally, quantifying the recognition rate achieved by experienced people working with cats is part of our future work.

- on line 344, the text now is

“3. quantifying the recognition rate achieved by experienced people working with cats when only the acoustic emission is available.”

3) l. 90 “hypothesized” not “supposed”

l. 133 “legitimate” instead of “legit”

l. 276 “were” not “where”

l. 278 “with respect to” rather than “wrt”

l. 281-283 SR, G and sc are not defined.

l. 290 “almost excellent” as noted in the previous review, this result IS excellent. I’d say “almost perfect”

l. 310 “first sight” not “first site”

Reference section has a mix of styles in terms of capitalizing words in titles. These should be consistent with the journal’s style.

Responses: Thank you very much for detecting these inconsistencies. To address, we made the following changes:

- on line 89, the text now is

“In presence of at least one veterinarian, cats were repeatedly exposed to three different contexts that were hypothesized to stimulate the emission of meows”

- on line 133, the text now is

“A posture change is considered a legitimate and realistic effect of the situation. Moreover, if the recording quality is sufficient to catch such an aspect, it can be considered as a useful additional information.”

- on line 279, the text now is

“…were determined by means of a grid search…”

- on line 280, the text now is

“The specific kernel was selected as it demonstrated superior performance with respect to other well-known ones, i.e., linear, polynomial homogeneous, polynomial inhomogeneous, and hyperbolic tangent.”

- on line 284, the text now is

“The parameters were taken from the following sets: spectral radius SR∈{0.8, 0.9, 0.95, 0.99}, reservoir size G∈{100, 500, 1000, 5000, 10000, 20000}, and scaling factor sc∈{0.1, 0.5, 0.7, 0.95, 0.99}.”

- on line 294, the text now is

“As we can see, the DAG-based classification scheme provides almost perfect recognition rate outperforming the rest of the approaches.”

- on line 314, the text now is

“At first sight,…”

Finally, all references now follow the MDPI format as defined in Latex.